# Does the Number of Reasons for Seeking Care and Self-Rated Health Predict Sick Leave during the Following 12 Months? A Prospective, Longitudinal Study in Swedish Primary Health Care

**DOI:** 10.3390/ijerph19010354

**Published:** 2021-12-30

**Authors:** Kristin Lork, Kristina Holmgren, Jenny Hultqvist

**Affiliations:** Department of Health and Rehabilitation, Institute of Neuroscience and Physiology, Sahlgrenska Academy, University of Gothenburg, 405 30 Gothenburg, Sweden; kristina.holmgren@neuro.gu.se (K.H.); jenny.hultqvist@gu.se (J.H.)

**Keywords:** self-perceived health, symptoms, complaints, sickness absence, primary care

## Abstract

Background: Sick leave has major social and economic consequences for both individuals and society. Primary Health Care (PHC) meets people who seek care before they risk going on sick leave. This study examined the impact of self-perceived health on sick leave within 12 months for workers seeking care in PHC. Methods: The study had a prospective longitudinal design with 271 employed, non-sick-listed patients aged 18–64 years seeking care for physical and/or mental symptoms at PHC. In a logistic regression, an estimation of the odds ratio (OR) for belonging to the group workers with >14 days of sick-leave (W-SL) was made. Results: A high number of reasons when seeking care, with an OR of 1.33 (confidence interval 1.14 to 1.56), and lower self-rated health, with an OR of 1.45 (confidence interval 1.10 to 1.91), were determinants for sick leave at 12 months after adjusting for covariates and confounders. Mental symptoms constituted the main reason for seeking care, followed by musculoskeletal pain, and significant differences in proportions regarding most symptoms were shown between the groups with and without sick-leave >14 days. Conclusion: Health care professionals in PHC need to be aware of the risk of future sick leave at comorbidity and low self-perceived health. Preventive rehabilitation interventions should be offered to improve health and prevent sick leave for this group.

## 1. Introduction

### 1.1. Background

Working is the foundation of human well-being and survival. For most people, work is beneficial for health, but shortcomings concerning the organisation of the workplace and the physical and psychosocial work environment may lead to illness or injury [1]. Illness and injuries occur for many reasons other than being work-related; however, they may lead to shorter or longer sick leave [2]. Sick leave has a large social and economic impact on both the employee and society [3,4]. Reality is complex, and several factors influence the process of working, going on sick leave, and returning to work again [5]. Perceived health is among the factors that influence sick leave, and the present study investigated the impact of perceived health on future sick leave in workers seeking care in primary health care (PHC). 

In Sweden, PHC meets the population’s needs for basic medical treatment, prevention, and rehabilitation that do not require the medical and technical resources of hospitals. Health care delivery is based on fundamental principles regarding being person-centred, with high availability and continuity and in collaboration with society at large [6]. Almost 70% of the Swedish population visits primary health care every year with a wide range of diagnoses, including physical and mental disorders [7]. Musculoskeletal disorders (MSDs) such as injuries and disorders affecting, for example, muscles, tendons, ligaments, nerves, and discs, and common mental disorders (CMDs), such as depression, anxiety disorders, and exhaustion disorder, are the leading causes for sick leave in Sweden today [8], and these disorders are often associated with work [3,9]. Perceived stress may result in both physical and mental symptoms such as headaches or muscle pain, weakened body immune system, cardiovascular diseases, insomnia, cognitive and emotional symptoms, and ultimately exhaustion disorder [10]. 

Most people with CMD work despite their symptoms, and they often struggle day by day to cope at work [11,12]. The symptoms may vary significantly over time [13], and people with CMDs risk deteriorating in function, which eventually may lead to sick leave [14]. People with work-related stress may seek care at an early stage at PHC centres to receive help for various physical and mental symptoms [15,16]. Research has shown that, at this early stage, both individuals and general practitioners (GPs) may be unaware that the symptoms are stress-related [16,17]. At first glance, people with stress-related symptoms may seem relatively healthy given that they are of working age, have gainful employment, and are not on sick leave. A study in Sweden [18] with a randomised controlled design (RCT) evaluated an intervention with systematic early identification of work-related stress among employed non-sick-listed women and men. The results of the RCT showed that about one-third of the study population had been on sick leave for more than 14 days at a 12-month follow-up [18]. This is considerably higher compared to the general working population, where the prevalence of sick leave is approximately 9% [19]. As there are associations between MSD and CMD in general, an individual may present several of these symptoms when seeking care [15,20]. The most frequent reasons for seeking care during the two years preceding sick leave for exhaustion disorder, reported in a Swedish study, were infections and anxiety/depression [17]. Furthermore, a majority had a comorbidity with other physical and mental symptoms, and the mean number of symptoms was 3.7 (SD 2.2). A German general population survey [21] showed, in a similar way, that about 9% of the population experienced somatic, anxious, or depressive syndromes with a comorbidity in one-third of the cases. Another German study [22] revealed that a high proportion of people with depression only reported physical symptoms when seeking care at PHCCs. They also reported twice as many complaints, including both physical and mental symptoms, with a mean of 2.02 (SD 1.33), compared to people without depression. This suggests that the number of health-related symptoms could be an indicator of health to consider when investigating predictors for sick leave among non-sick-listed, employed people seeking PHC.

Self-rated health (SRH) measured by the first question of the Short Form Health Survey (SF-36) [23] is a well-established method in research, and SF-36 is regularly used in health surveys as it captures physical, mental, and social dimensions of health [24]. A study among U.S workers [25] shows that a psychosocial work environment is associated with fair/poor SRH. Women usually rate their health as slightly lower than men in a general European population [26]. However, a Swedish study [27] showed that men have a poorer SRH than women in a population with just employees. There is extensive research on the risk factors for sickness absence [28,29,30,31], and poor self-rated health is one of the risk factors, as it is clearly associated with sick leave [32]. The single-item question on SRH may also predict a return to work better than more time-consuming questionnaires [33]. Even so, there are few studies investigating the self-rated health of people of working age seeking care early in PHC for various physical and mental symptoms that eventually may result in sick leave. 

With a broad mission to meet people’s basic needs for medical care, PHC has a challenge in identifying workers with multiple symptoms and poor health who may risk sick leave. To identify the risk of sick leave in an early phase and to offer adequate intervention, it is important to learn more about workers seeking care for different physical and mental symptoms. It is of interest to examine what they are seeking care for, the number of reasons for seeing care, and how they estimate their health at the time of consultation in PHC. The individual’s perspective is essential, and more knowledge is required to gain a deeper understanding.

### 1.2. Aim

The overall purpose of this study was to investigate if reasons for seeking care and SRH can predict registered sick leave (>14 days) at 12 months follow-up in non-sick-listed, employed women and men seeking PHCCs for physical and/or mental symptoms.

The specific hypotheses in the study are:There is a baseline difference in reasons for seeking care between those with and those without registered sick leave (>14 days) at 12 months follow-up.The number of reasons for seeking care at baseline predict registered sick leave at 12 months follow-up.Lower SRH at baseline predicts registered sick leave at 12 months follow up.

## 2. Materials and Methods

### 2.1. Study Design and Procedure

The present study is a prospective, longitudinal study, and it is a part of the TIDAS project, a two-armed RCT, Trial registration: ClinicalTrials.gov. Identifier: NCT02480855. Registered 20 May 2015 [34]. The overall aim of TIDAS was to evaluate whether the systematic early identification of work-related stress could prevent sickness absence. Results from the RCT are published elsewhere in several studies [18,35,36,37,38,39]. In a study protocol, Holmgren et al. presented detailed information on TIDAS [34]. 

Registered sick leave data were obtained from the Swedish Social Insurance Agency (SSIA), and the registration and recruitment of patients were conducted from May 2015 to January 2016. In Sweden, the employee is compensated by the employer during the first 14 days of sick leave, and after that period, benefits are granted from SSIA.

Seven PHC centres located in Region Västra Götaland, Sweden, took part in the RCT study. Included were non-sick-listed employed patients between 18 and 64 years old seeking care at the PHC centre for physical and/or mental symptoms. Excluded were patients who had been on sick leave for seven days or more during the last month and patients with a full or part-time disability pension. The total study population consisted of 271 non-sick-listed employed women and men [18,34]. The present study used the total study population as the overall purpose was to investigate if reasons for seeking care and SRH could predict registered sick leave (>14 days) at 12 months follow-up regardless of participant allocation in the original RCT study.

### 2.2. Intervention

Together with treatment as usual, the intervention comprised an assessment with the Work Stress Questionnaire (WSQ) [40], designed to early identify people with work-related stress at risk for sick leave, feedback by the GP at consultation, and suggestions on possible measures [18,34].

The control group received treatment as usual at the GP consultation. The treatment could include medical investigations, diagnostics, and discussions about preventive and rehabilitating measures. To minimise the bias in the study, participants in the control group completed the WSQ after consultation [34]. 

### 2.3. Sample Size 

For the original RCT study, a power calculation (with a two-sided test, statistical significance of *p* < 0.05, and 80% power) showed that 135 participants were needed in each group in the RCT to identify at least a 15% difference concerning the number of sick-leave days from SSIA, (i.e. >14 days or more) 12 months after inclusion. The original study results [18] showed that there was no significant difference between the intervention and the control group regarding the number of sick-leave days at the 12-month follow-up. No additional power calculation was performed for the present study using the total study population.

### 2.4. Study Population

Self-reported baseline characteristics were collected by a questionnaire designed explicitly for the RCT study. The characteristics of the study population (*n* = 271) at baseline and at 12-month follow-up between workers without sick leave (W) and workers with >14 days sick leave (W-SL) are presented in Table 1. In the study population, most of the participants were married/cohabitant or in a relationship, 78%. A majority were female, 68%, and half of the study population were between 31 and 50 years old. A large majority of the study population, 89%, had an educational level from secondary school or university. Only 5% were self-employed, almost half of the study population had a private employer, and the other half had a public employer. The chi-square test was used to analyse the differences between W and W-SL, and the only significant difference between the two groups was educational level (*p* < 0.038).

### 2.5. Measures

The outcome measure was sick leave when granted from SSIA (i.e. >14 days) at 12 months. Gross sick-leave days (>14 sick-leave days, irrespective of full- or part-time sick leave) during the 12 months after baseline were obtained from SSIA´s Micro Database for Analysing Social insurance (MIDAS). In a logistic regression, an estimation of the odds ratio (OR) for belonging to the group W-SL was made.

Data for the first exposure variable, the number of reasons for seeking care at baseline, were collected through the questionnaire designed explicitly for the RCT study. The question was, ‘What complaints are you seeking care for today?’ The participants had 15 response alternatives, and multiple reasons were possible. The response alternatives were mental symptoms such as fatigue, stress, sleeping problems, anxiety, depression, and other mental symptoms; musculoskeletal symptoms such as neck/shoulder pain, back pain, and other pain; other symptoms such as gastrointestinal symptoms, cardiovascular symptoms, skin symptoms/eczema/allergies, infections, accidents/injuries, or other symptoms. 

The second exposure variable was SRH. The question concerning self-rated health was obtained from the SF-36 [23]. The participants were asked, ‘How would you say your health is in general?’, rated on a five-point ordinal scale where 1 = excellent health, 2 = very good health, 3 = good health, 4 = fair health and 5 = poor health. This single item measure has been found to be a valid broad summary rating of health [41]. 

The socio-demographic data included factors such as sex, age groups (19–30, 31–50, 51–64), civil status (dichotomised as single or married/cohabitant/in a relationship), educational level (compulsory schooling, secondary school, university), occupational class (skilled/unskilled manual, medium/low nonmanual, high-level nonmanual), and employer (private employer, public employer, self-employed).

### 2.6. Statistical Analyses

The present study applied both descriptive and analytic statistics. Nonparametric statistics were used as the sick-leave data were largely skewed and as the variables were qualitative. 

The Mann–Whitney U test was used to test the differences regarding the first exposure variable, the number of reasons for seeking care, between W and W-SL. The differences in reasons for seeking care between W and W-SL in the population were investigated further by calculating the 95% confidence interval (C.I.) for the difference in proportions between W and W-SL on each reason for seeking care.

For the second exposure variable, SRH, the five-point ordinal scale was used in the logistic regression analyses. Socio-demographic variables were explored as covariates and confounders by means of the chi-square test and the Mann–Whitney U-test.

The outcome registered sick-leave days over 12 months was dichotomised into workers having no sick-leave days (W) (0) and workers having sick-leave days (W-SL) (1). Logistic regression models were then performed to study the influence of both exposure variables, adjusted for covariates, on registered sick leave.

In the logistic regression with both exposure variables, model 1 was unadjusted, model 2 adjusted for intervention/control, and model 3 adjusted for intervention/control plus educational level. The software used for the statistical analyses was IBM SPSS Statistics version 27. The level of statistical significance was set at *p* < 0.05 (95%).

### 2.7. Ethics Approval and Consent to Participate

Oral and written study information was given, and written consent was obtained from all participants, including consent for linking records to registers during follow-up. The Regional Ethical Review Board at the University of Gothenburg, Sweden, approved the project with the reference number 125-15. Additional approval was performed by the Swedish Ethical Review Authority with the reference number 2021–00627. 

## 3. Results

In the study population (*n* = 271), there were 96 (35%) W-SL (>14 days) during the 12 months following baseline.

### 3.1. Differences in Reasons for Seeking Care

In total, there were 15 reasons for seeking care, and multiple reasons were possible. The two major groups of reasons for seeking care were mental symptoms and musculoskeletal pain. Mental symptoms such as fatigue, stress, sleeping problems, anxiety, depression, and other mental symptoms were the main reasons for seeking care and constituted 55% of the reasons. Musculoskeletal pain was divided into neck/shoulder pain, back pain, and other pain and constituted 22% of the reasons. A 95% CI was calculated for the difference in proportions between W and W-SL on each reason for seeking care. There were significant differences in proportions regarding most mental symptoms such as fatigue, sleeping problems, anxiety, depression, and other mental symptoms. Symptoms of stress were the only nonsignificant difference in proportions between the two groups. Significant differences in proportions were also found for all symptoms of musculoskeletal pain, gastrointestinal symptoms, and cardiovascular symptoms. Detailed reasons for seeking care, with possible multiple answers, are presented in Table 2.

The participants were seeking care for a total of 638 reasons, with a median of one reason in the group of W and two reasons in the group of W-SL. The result of the Mann–Whitney analysis showed a significant difference (*p* < 0.001) regarding the number of reasons for seeking care between W and W-SL.

In the study population, 72% rated their health as good to excellent (Table 3). 

### 3.2. Number of Reasons When Seeking Care as a Determinant for Registered Sick Leave at 12 Months

Logistic regression analyses showed that a high number of reasons when seeking care were a determinant for sick leave in 12 months (*p* < 0.001) when adjusted for intervention/control and educational level. The adjusted odds ratio (OR) was 1.33 and the 95% CI was 1.14–1.56 (Table 4).

### 3.3. Self-Rated Health as a Determinant for Registered Sick Leave at 12 Months

Logistic regression analyses showed that lower SRH was a determinant for sick leave in 12 months (*p* = 0.008) when adjusted for the intervention/control group and educational level. The adjusted OR was 1.45 and the 95% confidence interval (CI) was 1.10–1.91 (Table 5).

## 4. Discussion

This prospective longitudinal study of non-sick-listed, employed women and men seeking PHC centres for physical and mental symptoms showed that a high number of reasons when seeking care and lower SRH were determinants for sick leave at 12 months, even after adjusting for possible covariates and confounders at baseline. The hypothesis that more health-related symptoms and lower SRH increase the risk for sick leave in 12 months was thus confirmed. Mental symptoms constituted the main reason for seeking care, followed by musculoskeletal pain. The results also discovered significant differences in proportions regarding most physical and mental symptoms when seeking care between the groups with and without sick leave >14 days during the following 12 months. 

### 4.1. Reasons for Seeking Care

The reasons for seeking care between W and W-SL showed significant differences regarding mental symptoms such as fatigue, sleeping problems, anxiety, and depression, and these mental symptoms were significantly more frequent in the group of W-SL. However, no significant difference between the groups was shown for stress. Although knowledge of the prospective associations between perceived stress and sick leave is still limited, this is a surprising finding. Some studies have shown an association between perceived stress and future sick leave. A Swedish study [16] among 198 women seeking care at PHC centres found that combined work-related and person-related perceived stress symptoms were associated with a higher risk of sick leave. A prospective Danish study [42] of almost 18,000 employees found an association between perceived stress and an increased risk of sick leave, particularly for women. In the present study, although all mental symptoms except perceived stress had significant differences between the groups, these symptoms frequently occur in connection with stress. A cross-sectional study [43] examining health-related factors associated with self-rated exhaustion disorder among public employees in Sweden found that cognitive problems, sleep problems, depressive symptoms, high stress, and stomach problems were associated with self-rated exhaustion disorder. A possible explanation in the present study regarding the nonsignificant difference between the groups concerning stress may be that some individuals did not relate their physical or mental symptoms with stress when seeking care, as they were unaware of the potential connections. Sleeping problems were a reason for seeking care to a greater extent for the W-SL in the present study. A systematic review [44] has shown that sleep disturbances alone are a risk factor for sick leave and increase the risk of sick leave by up to 23%. This is in line with the present study results, as there was a significant difference in proportions between W and W-SL regarding sleep problems.

Musculoskeletal pain constituted 22% of the reasons for seeking care, and various kinds of pain were the second leading reason for seeking care in the present study. There are often links between mental and physical symptoms, which is well recognised both from clinical experience and from research [45]. Pain, especially chronic pain (>3 months), often has multiple causes and leads to major psychosocial effects [46]. Psychosocial factors related to work shows that somatisation and low job security are risk factors for back pain [47]. Work-related stress may cause a variety of symptoms, and among the physical symptoms, chronic pain in the neck and shoulders [48] and back pain [45] are common. Chronic pain was associated both with high psychological job stress and physical stress in Japanese healthcare workers. Furthermore, chronic pain was also associated with depression [49]. Therefore, it is possible that some participants who sought care for musculoskeletal pain in the present study experienced psychosocial stress related to work or had mental symptoms that contributed to the pain experience. The biopsychosocial model [50], widely accepted in research concerning sick leave, can be useful in understanding the complex interactions that exist between physical and mental symptoms and the psychosocial environment. Therefore, a more holistic approach offered by the biopsychosocial model is suggested in PHC to comprehend the experience of health in a larger context.

### 4.2. Number of Reasons When Seeking Care

Comorbidity is common in primary care populations [15,16], and the number of reasons for seeking care may be seen as a form of self-assessment of health, which complements the measurement of SRH in the present study. Although most of the study population had few reasons for seeking care, there was a significant difference between the groups, where W had a median of one reason and W-SL had a median of two reasons. In the present study, multiple reasons for seeking care increased the risk of future sick leave, which is supported by studies displaying that about half of the patients who developed exhaustion disorder had a comorbidity with other mental or physical diseases [17,45]. 

As a provider of basic health care for the population, PHC has a key role in treating people with comorbidity. A vast majority of psychiatric consultations and treatments for persons with CMD occur at the PHC level, and 80% or more have contact only within PHC [51]. It is also at the PHC level that most sick leave certificates are issued for these symptoms [52]. For health care professionals working within PHC, it is important to include not only what symptoms the patient is seeking care for but also the number of symptoms and possible comorbidities in the assessment. However, this can be challenging as it requires time, the PHC is guided by a high demand for prompt access and care, and the workload is often high and may lead to chronic stress among the healthcare professionals [53]. The PHC services may also be challenged by insufficient resources and lack of support and collaboration between health care professionals [36]. Moreover, it can be difficult to get an overall picture of a person’s health status due to the large variation in reasons for seeking care and the complex relations that exist between physical and mental symptoms. Despite these complex conditions, at comorbidity, collaboration is needed internally at PHC centres. Collaboration in offering preventive rehabilitation measures to reduce the risk of future sick leave is also essential with other stakeholders in society.

### 4.3. Self-Rated Health

It is a surprising finding that more than one-third of the study population had a sick leave of more than 14 days the following year, significantly higher than the prevalence in the working population in Sweden [19]. This indicates that the study population was more vulnerable than the descriptive SRH displayed, as a large majority of the participants rated their health as good to excellent. However, the logistic regression models showed that lower SRH was a predictor for sick leave. This is consistent with other studies showing that poor SRH is a marker for increased length of sick leave [54] but also for increased risk of permanent work disability [55]. Similar findings were reported in a Belgian cohort study [56], based on a biopsychosocial model, that examined psychosocial factors predicting long-term sick leave. Fourteen factors showed significance for the length of sick leave, and among these were the patient’s perception of health. With better perceived health, the sick-leave period was shorter [56]. A large Dutch cohort study [31] had a similar purpose of investigating factors predicting long-term sick leave but was based on a broader multivariate risk model. It showed eleven factors that predicted sick leave, and one of these was poor self-rated physical health. Poor SRH is also a health-related factor associated with self-rated exhaustion disorder, which, in turn, increases the risk of long-term sick leave [43]. Results from a large Swedish longitudinal study [57] displayed in a similar way that self-rated health had significant associations with sick leave and disability pension. Moreover, the predictive power of SRH was strong and consistent.

Previous research has shown that SRH gives a general health assessment, including a large variety of health determinants [41]. The single-item question measuring SRH is easy to administer and provides a global approach to health. It has no reference to age or time, as there is no intention of measuring temporary health problems [41]. A qualitative focus group study [58] has shown that almost 80% of the participants considered physical health aspects when assessing their SRH. Moreover, 80% also included aspects beyond physical health such as functional, coping, and well-being dimensions. Other studies displayed that SRH not only reflects the person’s health status but also attitudes and norms in relation to health in groups and societies in general [59,60]. The predictive power of SRH in terms of sick leave, as the present study demonstrates, in line with previous research, may be related to the global approach and the multidimensional concept behind the SRH. Within PHC, the question of SRH can be used as a tool for quick assessment of the need for preventive measures. For people with comorbidity, in particular, it may be a way to prevent sick leave.

### 4.4. Strengths and Limitations of the Study 

There are strengths and limitations with the present study that need to be highlighted. The results need to be interpreted based on the contextual limitations the study provides. Some of the limitations are as follows. The study population was small, and the findings must be interpreted with caution as they might not be valid in future studies with larger study populations. Even so, the present study is unique by examining a relatively healthy population seeking primary health care before they risk being on sick leave. Comorbidity, manifested as multiple reasons for seeking care in the present study, and SRH are associated with sick leave in several studies [20,31,54,57], strengthening this study’s results. SRH is a subjective assessment of health and may, therefore, both be overestimated or underestimated in comparison to other assessments, and as with any self-rating question, the answer may be affected by bias such as social desirability. Even so, SRH has proved to be a valid predictor for mortality and sick leave in previous research [54,57]. The present study was part of a larger research project, and only research questions fitting the data within the original study’s framework could be processed. Other potential variables for self-perceived capacities, such as self-perceived work capacity, in relation to SRH and future sick leave, might have been interesting to investigate. 

A strength of the present study was the prospective longitudinal design, making it possible to investigate the temporal mechanisms between self-perceived health and sick leave and to make predictions about the associations. Another strength of the study was the availability of registry data on sick leave. Registry data minimised the risk of dropouts, and there was no risk of recall bias. The groups of W and W-SL were also similar in terms of sociodemographic factors, which facilitated comparisons between the groups. The only significant difference between the groups was educational level. 

To our knowledge, no previous studies have examined the risk of future sick leave in a working population based on self-perceived health in a PHC context. The present study shows associations between perceived health and the risk of future sick leave at an individual level. However, several factors affecting sick leave exist, in addition to health. Sick leave is a complex phenomenon that needs to be understood in a larger context, including individual characteristics, work environment, socio-economic factors, and societal factors [5]. In this complex context, the assessment of comorbidity and SRH can be a useful part of predicting the risk of sick leave.

## 5. Conclusions

The findings from the present study indicate the importance of making an overall assessment of health that includes comorbidity and SRH. It is essential for health care professionals in PHC settings to be aware of the risk of future sick leave at comorbidity and low self-perceived health. To meet the challenge of offering person-centred care with good availability and continuity within the PHC, an organisation is needed that facilitates cooperation and communication between the patient and the healthcare professionals. A functioning collaboration within the PHC and collaboration with other stakeholders are also crucial. In the case of comorbidity and low self-perceived health, collaboration may facilitate preventive rehabilitation measures adapted to the person’s preferences and needs to improve health and reduce the risk of future sick leave. Future research should develop and evaluate methods supporting collaboration between the PHC and other stakeholders on a societal level.

## Figures and Tables

**Table 1 ijerph-19-00354-t001:** Baseline characteristics of the study population, *n* = 271, and workers without (W) and with sick leave (W-SL) at 12-months follow-up.

Characteristics	Total Population at Baseline *n* = 271 (%)	W at 12 Month *n* = 175 (%)	W-SL at 12 Month *n* = 96 (%)
Sex			
Female	185 (68)	114 (65)	71 (74)
Men	86 (32)	61 (35)	25 (26)
Age groups			
19–30 years	47 (17)	32 (18)	15 (15)
31–50 years	134 (50)	88 (50)	46 (49)
51–64 years	90 (33)	55 (32)	35 (36)
Birthplace ^1^			
Nordic countries	247 (91)	159 (91)	88 (92)
Other	23 (9)	15 (9)	8 (8)
Civil status ^1^			
Single	58 (22)	33 (20)	25 (26)
Married/cohabitant	197 (73)	131 (75)	66 (69)
In a relationship	14 (5)	9 (5)	5 (5)
Educational level ^1^			
Compulsory schooling	28 (11)	**16 (9)**	**12 (12)**
Secondary school	120 (44)	**76 (44)**	**44 (46)**
University or higher	122 (45)	**82 (47)**	**40 (42)**
Occupational class ^2^			
Skilled/unskilled manual	107 (40)	62 (35)	45 (47)
Medium/low non-manual	114 (42)	76 (44)	40 (42)
High-level non-manual	47 (18)	36 (21)	11 (11)
Employer ^1^			
Private	131 (49)	84 (49)	42 (46)
Public	125 (46)	76 (44)	49 (51)
Self-employed	14 (5)	12 (7)	2 (3)

Differences between W and W-SL were analysed using chi-square. Bold numbers indicate a significant difference. Note: ^1^ One missing value. ^2^ Three missing values.

**Table 2 ijerph-19-00354-t002:** Reasons for seeking care, with possible multiple answers, in the total population, *n* = 271, and in the group of workers (W), *n* = 175, and the group of workers with sick leave (W-SL), *n* = 96.

Reasons for Seeking Care	Total Study Population *n* = 271 (%)	W *n* = 175 (%)	W-SL *n* = 96 (%)	Confidence Interval between W and W-SL ^1^
Fatigue	92 (34)	45 (26)	47 (49)	**−0.352–0.113**
Stress	77 (28))	44 (25)	33 (34)	−0.207–0.022
Sleeping problems	68 (25)	32 (18)	36 (37)	**−0.305–0.08**
Anxiety	64 (24)	32 (18)	32 (33)	**−0.261–0.04**
Gastrointestinal symptoms	54 (20)	16 (9)	38 (40)	**−0.411–0.198**
Musculoskeletal pain, other	51 (19)	18 (10)	33 (34)	**−0.346–0.136**
Musculoskeletal pain in neck/shoulder	50 (18)	24 (14)	26 (27)	**−0.236–0.031**
Depression	44 (16)	22 (13)	22 (23)	**−0.201–0.006**
Other symptoms	41 (15)	13 (7)	28 (29)	**−0.316–0.119**
Musculoskeletal pain, back pain	37 (14)	15 (9)	22 (23)	**−0.237–0.05**
Cardiovascular symptoms	32 (12)	11 (6)	21 (22)	**−0.246–0.066**
Mental symptoms, other	10 (4)	3 (2)	7 (7)	**−0.111**–**0.000**
Skin symptoms, eczema, allergies	10 (4)	6 (3)	4 (4)	−0.056–0.041
Infections	5 (2)	0 (0)	5 (5)	
Accident or injury	3 (1)	1 (0.5)	2 (2)	−0.046−0.016
Total	638	282	356	

Note: ^1^ The 95% confidence interval for proportions regarding the reasons for seeking care is indicated between W and W-SL. Bold numbers indicate a significant difference in proportions between W and W-SL.

**Table 3 ijerph-19-00354-t003:** Self-rated health at baseline, *n* = 258 ^1^ for the total study population.

	*n* (%)					
	Excellent	Very Good	Good	Fair	Poor	Total
Total	18 (7)	59 (23)	108 (42)	60 (23)	13 (5)	258 (100)

Note: ^1^ 13 missing values.

**Table 4 ijerph-19-00354-t004:** The association between high number of reasons for seeking care and sick leave in 12 months.

	Model 1, OR (95% CI)	Model 2, OR (95% CI)	Model 3, OR (95% CI)
High number of reasons for seeking care	1.31 (1.13−1.52)	1.33 (1.14−1.55)	1.33 (1.14−1.56)

Model 1 unadjusted OR, Model 2 adjusted OR for intervention/control, Model 3 adjusted OR for intervention/control plus educational level. 95% CI, confidence interval; OR, odds ratio.

**Table 5 ijerph-19-00354-t005:** The association between lower self-rated health and sick leave in 12 months.

	Model 1, OR (95% CI)	Model 2, OR (95% CI)	Model 3, OR (95% CI)
Lower self-rated health	1.45 (1.10–1.90)	1.46 (1.11–1.92)	1.45 (1.10–1.91)

Model 1 unadjusted OR, Model 2 adjusted OR for intervention/control, Model 3 adjusted OR for intervention/control plus educational level. 95% CI, confidence interval; OR, odds ratio.

## Data Availability

The data presented in this study are available on request from the corresponding author. The data are not publicly available, due to ethical reasons.

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
