# Peer review of "Does the Number of Reasons for Seeking Care and Self-Rated Health Predict Sick Leave during the Following 12 Months? A Prospective, Longitudinal Study in Swedish Primary Health Care"

_ijerph, 2021, doi:10.3390/ijerph19010354_

Round 1

Reviewer 1 Report

The paper investigates a research question that is of clear relevance for readers of IJERPH. Does the number of reasons for seeking care and self-rated health predict sick leave during the following 12 months? The analysis relies on data from 271 employed, non-sick-listed patients, aged 18-64 years, seeking care for physical and/or mental symptoms at the Primary Health Care.

The paper is very well written with well-defined research questions. The research methodology is presented and executed competently, and the results from the analysis are presented in a clear and insightful way. This is a well-executed research contribution. I recommend publication in IJERPH subject only to normal copy editing.

Author Response

Thank you for your kind review and the recommendation for publication!

Reviewer 2 Report

This study is important and interesting. The individual perspective is essential to prevent sick leave and also to find evidence for rehabilitation measures  to prevent long-term sick leave. 

The manuscript is well written and only small corrections are needed.

Author Response

Thank you for valuable comments that may improve the study and make the results clearer. Attached is a review, point by point, of the comments we received from you.

Reviewer 3 Report

The introduction gives a convincing motivation for why the topic is important in general. However, I don't fully see yet, why the fact that number of reasons for visiting SRH is correlated with the probability for sick leave, is important. Further, how is it surprising that people, who feel less healthy are more likely to have a sick leave compared to people who feel rather healthy? I'm afraid I must have missed something. I'm sure that there are reasons for why that study is relevant.

I expect from a scientific paper that a hypothesis is not just stated (here one sentence on page 3) but derived. There are two common ways of how a hypothesis can be derived. Option 1: The authors write a theory that leads to predictions. Option 2: The authors derive their theory from previous findings in related literature. (Option 3: A combination of Option 1 & 2.) Please improve that. Furthermore, there are three research questions. Thus, there must be (at least) three clearly stated (and derived) hypotheses.

To be honest, I’m confused about what is compared to what. From the description of the intervention (treatment group and control group) it seemed natural to me to assume that the study is meant to compare these two groups. If that is the case, please consider the following argument:

  • From the practical point of view I understand that it’s interesting to see, whether the complete intervention has an effect. From the scientific point of view, however, I wish one could disentangle, which of the components of the intervention lead to the effect. One could argue that, for example, the 2 h GP training in the use of the WSQ already is enough to make people more sensitive to that topic and thereby that alone could have led to an effect. I see why it’s impossible to rerun the study with several treatments groups that vary the changes one-by-one. But at least that issue should be discussed and a disclaimer should be provided.

From looking at the research questions and the sentence in line 163 (“present study used the total study population,”) plus from looking at the data it seems as if no difference between these two groups has been made. If the latter is true, please resolve that confusion in the paper. For that, section “2.2 Intervention” would need to be re-written.

Was there no interaction between the treatment groups and the effect of the explanatory variables on the outcome measures? It would be interesting to see that.

I would perceive it as helpful if there were some graphs included that visually highlight the findings of the study. I would also appreciate to see tables of the models to be able to judge whether the results are valid. 

Given the current presentation of the data I cannot fully judge, whether the data really support the conclusion that have been derived in the paper.

Author Response

Thank you for valuable comments that may improve the study and make the results clearer. Attached is a review, point by point, of the comments we received.

Round 2

Reviewer 3 Report

I still don't see the value of that paper for the scientific commuity. For example, the authors simply state in their response letter that they have derived their hypotheses from literature, which is simply not the case. However, it seems that the journal is OK with such a quality of papers and that a sugestion for a rejection by a referee is interpreted as a suggestion for an R&R. It seems as if I'm unable to adjust my expectations for a scientific paper to the standards expected in that journal. Therefory, I decided to just suggest acception of the paper and will no longer invest my time and effort in trying to carefully review papers for that journal.